# The Mediation Chain Effect of Cognitive Crafting and Personal Resources on the Relationship between Role Ambiguity and Dentists’ Emotional Exhaustion

**DOI:** 10.3390/ijerph192416617

**Published:** 2022-12-10

**Authors:** Rosana Stan, Cristina Ciobanu

**Affiliations:** 1Department of Psychology, University of Oradea, 410087 Oradea, Romania; 2Department of Dental Medicine, University of Oradea, 410087 Oradea, Romania

**Keywords:** role ambiguity, burnout, cognitive crafting, personal resources, dentists

## Abstract

Based on Job Demands-Resources (JD-R), Conservation of Resources (COR) and self-regulation theories integration, this study investigated the association between role ambiguity and emotional exhaustion among 191 Romanian dentists, as well as the chain mediating role of cognitive crafting and three personal resources (resilience, optimism, and self-efficacy). Three conceptual models which included, separately, the three personal resources were proposed. PROCESS macros were used to verify the hypotheses related to the testing of the path mediation models. The results indicated that role ambiguity was directly and positively associated with dentists’ burnout. More importantly, the sequential indirect effect of role ambiguity on burnout via mediators in chains (cognitive crafting and resilience for the first model; cognitive crafting and optimism for the second model; cognitive crafting and self-efficacy for the third model) was significant. The findings provide a direction for dentists’ health intervention because it reveals how the negative impact of role ambiguity on emotional exhaustion increasing can be buffered by the cumulative effect of cognitive crafting and different personal resources, as a result of their chain reinforcement.

## 1. Introduction

Job burnout as an enduring psychological condition of ill-being when employees are no longer able to invest effort in their work [1] has already been shown to negatively affect healthcare professionals at higher rates [2]. The quick change in job demands and the depletion of resources as consequences of the COVID-19 pandemic [3,4] seem to have accelerated the burnout syndrome among medical staff. When dentists burn out from their jobs, they are prone to serious risk of negative mental and physical consequences, such as work-related anxiety and depression [5]. Moreover, increased risk of cardiovascular diseases or Type 2 diabetes were also reported [6]. Burned-out physicians are more likely to deliver suboptimal patient care or make medical errors [7]. Since the repercussions of this syndrome are embodied in clinical errors [8], the reduction of medical staffs’ burnout becomes essential. For dentists, the targeted professional group in the present research, a recent meta-analysis [9] concluded that there was a considerable prevalence of burnout, mainly in the subscale of emotional exhaustion, a key dimension as the first stage of the syndrome [10]. Moreover, for Finnish dentists, a longitudinal study showed that burnout predicted depressive symptoms and life dissatisfaction [11]. Based on the arguments presented above, the question that arises is how dentists can avoid becoming exhausted by their profession?

Research over past decades revealed strong support for the JD-R model [12] in explaining the appearance of burnout. As the leading framework for the current research, JD-R theory [12] states that the combination of high job demands and low job resources triggers burnout appearance, as a health impairment process [12,13]. Role ambiguity, the focus of the current research, is a specific job demand in JD-R theory. Acknowledged to be the most important job demand next to role conflict, role stress, stressful events, workload, and work pressure [14], role ambiguity plays a crucial role in burnout prediction [15]. On the other hand, job crafting, an adaptive self-regulation strategy, when employees proactively optimize the work environment by adjusting their tasks, job demands, or job resources [1] usually results in new personal and job resources, because it modifies the stress-response or the stressor [1]. The JD–R model acknowledges the importance of job crafting for preventing burnout [16] and most of the available evidence suggests that job crafting leads to lower levels of burnout [17] by significantly decreasing exhaustion levels of employees [18,19]. Considering that resources are defined as anything perceived by individuals to help achieve their goals [20], we approach cognitive crafting as another type of resource which negatively relates to burnout. Existing research has identified various antecedents of job crafting, such as job demands [21], as well as the fact that job crafting results in various proximal outcomes, such as positive feelings and attitudes [22]. This means that job crafting should be instrumental in reducing role stress through proactive initiatives taken to increase one’s resources [23], an assumption which is central in the Conservation of Resources theory (COR, [24]). Personal resources, such as resilience, optimism, and self-efficacy, are another type of resources understood as personal characteristics which an individual can mobilize in stressful situations [1]. An important extension of the original JD–R theory [25,26] is the inclusion of, and the call for, consideration of the role of personal resources in the relationship between job demands, job resources and different outcomes [1]. Moreover, next to situational factors, like work overload, individual factors (e.g., self-efficacy) were classified as antecedents of burnout [27,28]. Personal resources, just like job resources, are motivational because they help employees reach their work-related goals [1]. Nevertheless, the COR theory [29], which complements the JD-R theory in explaining burnout [30,31], emphasizes the importance of resources in avoiding burnout [23]. Thus, although important in explaining burnout through the prism of a combination of high demands and low job resources [32], we chose, for the current research, not to focus on job resources in explaining the relationship between role ambiguity and dentists’ burnout, but to rather focus on the mediating role of individual factors, like cognitive crafting, resilience, optimism, and self-efficacy. Based on the COR theory [29] we assumed that the call to job crafting would help dentists gain personal resources [33] and, further, through a chain reinforcement, burnout could be counteracted, because the burdens of professional work might be balanced by these personal resources. Thus, we expected that cognitive crafting would reinforce resilience, self-efficacy, and optimism in a mediation chain through the effect of the resource caravan and this mediation chain would buffer the impact of role ambiguity on burnout. Overall, there is a dearth of studies that have focused on examining how job crafting is resourceful in decreasing burnout. The impact mechanism of the interrelated factors is unclear and needs further exploration. To cover this gap, and based on the above-mentioned arguments, the goal of this study was to clarify the relationship between role ambiguity, as job demand, and dentists’ burnout and to examine the role of cognitive crafting and personal resources in reducing dentists’ burnout. By integrating assumptions from three theories (i.e., JD-R, COR, and self-regulation theories) in conceptualization of the models, the current research approached cognitive crafting as the mediator between role ambiguity, as job demand, and burnout, and explored the increase of personal resource availability when they are included in a mediation chain. To our knowledge, this relationship has not been tested before. We proposed and analyzed three mediation models with three paths each. The cognitive crafting was included as the first mediator (M1) for all the three models, and each personal resource as the second mediator (M2) (resilience for the first model, optimism for the second model, and self-efficacy for the third model).

In sum, this article partially appeals to JD-R theory for analyzing the relationship between dentists’ role ambiguity and burnout but integrates self-regulation perspectives and COR theory into JD-R theory, gaining an approach from a multifaceted perspective and contributing to the research literature in the following ways. First, it brings the unique perspective of the mediator role in the relationship between role ambiguity and burnout to the fore, allowing exploration of an underlying mechanism unaddressed so far in this formula. Thus, the study contributes to the burnout literature by suggesting the effectiveness of job crafting in reducing the role ambiguity on burnout. Second, following the concept of resource caravans in COR theory, this study suggests that job crafting results in higher individual resourcefulness, as evident through multiple mediation effects, which, in turn, buffers the negative effect of role ambiguity on burnout. The results have practical implications because they might serve to assist in the designing of efficient intervention programs. By trying to answer the practical research question of how dentists can avoid becoming exhausted by their profession, we believe that our models could be a first step in identifying strategies to prevent and treat this issue.

### 1.1. Theoretical Background and Hypotheses

#### 1.1.1. Burnout and Role Ambiguity in JD-R Theory

Occupational burnout is a state of physical and emotional exhaustion caused by excessive tasks imposed by the physical or social work environment [34], a long-term reaction to chronic stress related to work [35]. According to the JD-R model of burnout [26,36], job demands and job resources, two categories of elements we encounter at work, might, in certain conditions, represent risk factors associated with job stress [12]. Thus, the JD-R model [26] explains how attempting to meet job requirements involves effort directed at maintaining the desired level of performance. If not balanced with an adequate set of resources, this effort produces psychophysiological costs, which can lead to burnout symptoms [26]. The empirical evidence of the JD-R model for the understanding of the triggering mechanisms of burnout is sustained by a large body of studies (i.e., [16,37,38,39]). In professions such as mental health, healthcare, social services, education or justice, occupational burnout frequently affects workers because these professionals face strong emotional experiences at work as a job characteristic [35]. Burnout syndrome occurs gradually and increases progressively in severity through its three dimensions [40] which are, according to Maslach and Leiter [41], emotional exhaustion (general fatigue, lack of energy, lack of enjoyment of life, and irritability), depersonalization (a negative or excessively detached response toward the service, lack of empathy towards service recipients) and reduced personal accomplishment (decline in one’s feelings of personal achievement and competence). Thus, in the first phase called emotional exhaustion, employees are overwhelmed in terms of emotional resources [35] and report consistent feelings of tiredness, and chronic fatigue [42]. As many studies showed, physicians have a higher prevalence of burnout than the general population and any other profession [43,44,45,46]. Further, it has been suggested that dentistry particularly generates more stress than any other profession and that job-related factors explain almost half of the overall stress in a dentist’s life [47]. Previous research already showed that the dental profession is generally linked to high levels of burnout syndrome [48,49,50] mainly in the subscale of emotional exhaustion [9,51]. When referring to the burnout division into three dimensions (emotional exhaustion, depersonalization, and reduced personal accomplishment/professional efficacy), emotional exhaustion is viewed as the core component of burnout, while depersonalization and reduced personal accomplishment are additional components [28,52]. It is argued that exhaustion, as a form of strain, has a separate role from cynicism and personal accomplishment in burnout [14]. While cynicism is seen as a form of defensive coping in the burnout phenomenon, personal accomplishment is considered a form of self-evaluation related to performance [14,53], reflecting the employee’s personality and not his or her reactions to stressful situations [54]. Moreover, in defining burnout, Pines and collaborators [55] refer to the state of emotional, but also physical and mental exhaustion in long-lasting emotionally demanding situations. Consequently, it was argued that emotional exhaustion was especially apparent as the core component of burnout [56]. Besides, a systematic review pointed to a relationship between psycho-social working conditions and the development of emotional exhaustion as the core dimension of burnout [56]. However, research conducted on Romanian healthcare professionals measured emotional exhaustion as the representative component of burnout [57]. Therefore, for the present study we decided to measure emotional exhaustion as a single dimension of burnout if the manifestations like constant fatigue or lacking strength were the common elements for every dentist as the first phase of burnout syndrome [35]. Furthermore, among other job demands, role ambiguity is a significant variable for the energy-driven process in the JD-R model [15,16,26] which explains much of the variants in burnout, specifically in the form of depersonalization and exhaustion [58,59]. Described as a term which refers to” the lack of clarity, certainty and/or predictability one might have expected with regards to behavior in a job” [60], role ambiguity is positively associated with burnout and its subcomponents [15,61]. Role ambiguity triggers an affective state which includes anxiety, depression, lack of self-confidence, or dysfunction in dealing with social situations [14,62]. The presence of role ambiguity among dentists might be explained by the common triggering mechanism affecting employees when they do not understand the goals they should be pursuing (with difficult clients) and what their priorities should be [60]. Previous research examined the relationship between role ambiguity and burnout in the case of healthcare professionals [63,64] but, to our knowledge, this relationship was not tested in the case of dentists. Consequently, and based on prior findings, we proposed the following research hypotheses:

**Hypothesis** **1** **(H1).***Role ambiguity is positively related to burnout*.

#### 1.1.2. Personal Resources and Cognitive Crafting in the Relationship with Burnout

It has been stated that it is not always possible to predict the intensity of the symptoms associated with burnout [65] because individual variables can act as protective factors [66,67]. Individual factors refer to individual differences or personal characteristics that are relatively stable over situations and time [16]. In the context of the JD-R theory, personal resources are understood as the personal aspects that an individual can mobilize to cope with job demands to reduce the chances of job strain, such as burnout [1]. Personal resources may include variables, such as self-efficacy or optimism [68], which are expected to function as a buffer between high job demands and undesirable health consequences [12]. Resilience, on the other side, has been widely recognized and studied as a personal resource over the years [69,70,71], frequently targeting healthcare workers (i.e., [72]) and the role of resilience in reducing burnout (i.e., [73]). Previous research demonstrated that positive psychological resources, such as self-efficacy and optimism, counteract the distress from job demands [74] which, in turn, leads to lower levels of burnout. In addition, although there are four recognized positive psychological resources (hope, optimism, efficacy, and resilience) which, when combined, emerge in a second-order core construct, referred to as psychological capital [74,75] these resources were empirically demonstrated to be discriminant constructs [75]. Moreover, considerable research on each of the individual components of psychological capital was conducted because each component of psychological capital is desirable in an organization [76]. The current research focused on optimism, efficacy, and resilience and analyzed them separately as the second link in the mediation chain, after cognitive crafting. According to the theory of self-regulation, individuals use self-regulation resources to straighten the difference between a current state and a desired state [77]. As an individual variable, job crafting indicates the extent to which individuals change their behavior according to their abilities, by appealing to different self-regulating behaviors [78], to balance work requirements and resources [21], which decreases burnout [17,18,19]. As a reference point for the present research, the results of the cross-lagged longitudinal structural equation modelling demonstrated that job crafting predicted psychological capital over time and no reverse causation effects were found [79]. Consequently, we decided to include cognitive crafting as the first mediator in the mediation models. Based on the arguments presented above, the research question that arose was:” Which personal resources buffer the impact of job demands, like role ambiguity, when added in the mediation chain after cognitive crafting on dentists’ burnout?” As mediating variables in the relationship between role ambiguity and burnout, resilience, optimism, self-efficacy, and cognitive crafting are described as follows.

#### 1.1.3. Resilience

APA [80] defines resilience as “the process of adapting well in the face of adversity, trauma, tragedy, threats, or even significant sources of stress”, but it was noticed that there are profound differences in how people cope with stressful experiences [81]. It was proposed that a well-functioning mapping system, which serves to integrate information from different sources, like the current situation or prior experience, as well as more conscious and goal-driven processes, is an integral part of the cognitive basis for resilience to adversity [81]. However, when it is enhanced, resilience is an important resource needed to process and overcome hardship, which further promotes well-being behaviors [82,83] and reduces general burnout levels [84,85]. Recent research, focusing on doctors [85,86,87,88], confirmed the negative relationship between resilience and burnout. Resilience had a moderation effect on burnout in the case of nurses working at hospital centers [89] and higher levels of resilience were associated with lower levels of emotional exhaustion for healthcare workers during the second wave of COVID-19 in Portugal [73]. Psychological research demonstrates that the resources and skills associated with more positive adaptation (i.e., greater resilience) can be cultivated and practiced [90]. Based on the arguments presented above, resilience should be considered a valuable resource for reducing burnout, and, consequently, we formulated:

**Hypothesis** **2** **(H2).***Resilience is negatively related to burnout*.

#### 1.1.4. Optimism

Optimism is a positive future expectation open to development [91], the attitude that good things happen and that wishes or aims will ultimately be fulfilled [80]. Optimism is also defined as a positive attribution to succeeding now and in the future [75]. Furthermore, optimism was described as an explanatory style interpreting negative events as external, temporary, and situation specific, and positive events as having opposite causes (i.e., personal, permanent, and pervasive) [92]. Through an adaptive management of personal goals, and by using active coping tactics [93], this psychological variable is commonly believed to be a protective factor regarding well-being and psychical health [94]. A path-analysis indicated that optimism remained a strong predictor of risk for job burnout, independent of stress [95]. Moreover, optimism was found to have the strongest relationship with the decreased personal accomplishment of burnout among hospital nurses [96]. A large body of research showed that optimism correlates with risk for job burnout [95,97,98,99] but, to our knowledge, the literature is scarce regarding the role of optimism in dentists’ burnout. Concluding that optimism could function as an important protective factor against dentists’ burnout, we formulated the following hypothesis:

**Hypothesis** **3** **(H3).***Optimism is negatively related to burnout*.

#### 1.1.5. Self-Efficacy

Defined as the belief in one’s ability to activate the cognitive resources, motivation, and courses of action necessary to meet the specific demands of a situation [100], self-efficacy functions as a buffer between high job demands and undesirable health consequences [27,28] because employees with a high sense of self-efficacy persistently pursue their goals and strongly believe that their actions will be successful [101]. Self-efficacy is studied as a personal resource that may protect one from the experience of job strain and, thus, make the escalation of burnout less likely [102]. The reason is that people with high self-efficacy view difficult obstacles as surmountable [103] and have confidence (self-efficacy) to take on, and put in, the necessary effort to succeed at challenging tasks [75]. Moreover, employees with high occupational self-efficacy generally worry less about work-related issues in their free time and find it easier to mentally detach and rest more effectively [104,105]. Apart from generating benefits like general health [106] or lower psychological distress [107], self-efficacy has been shown to have a negative association with emotional exhaustion in general surgery residents [108]. In a meta-analysis study, a significant self-efficacy–burnout relationship was observed across countries [109]. Research highlighted that self-efficacy had a negative effect not only on current burnout, but also on future burnout [102]. Yet, despite a large body of literature identified for other professional groups regarding the relationship between self-efficacy and burnout, in the case of dentists we found less in the way of relevant studies (i.e., [110]). Based on the from arguments presented above, we formulated:

**Hypothesis** **4** **(H4).***Self-efficacy is negatively related to burnout*.

#### 1.1.6. Cognitive Crafting

If working conditions are not favorable, employees may shape their jobs. This form of meaningful self-regulation [111] or bottom-up adjustments of demands and resources, seems to play a substantial role in the mechanisms suggested by JD–R theory [16]. The process is referred to as job crafting and is defined as the physical and cognitive changes individuals make in the work situation, but also in the general motivational state is related to the self [78]. A recent meta-analysis [112] showed that job strain (exhaustion and burnout) was negatively related to job crafting. Moreover, previous research emphasized that job-crafting interventions were associated with lower levels of exhaustion [113]. In recent years, researchers have developed theoretical frameworks for describing the complex work activities involved in job crafting [114,115,116]), but, for the present research, we followed the cognitive-behavioral perspective on job crafting [78,117] and chose to measure cognitive crafting. Cognitive crafting reflects cognitive adjustment at work when people reframe or redefine their job cognitively, forming it into a meaningful entity [78,118]. This adjustment can boost work motivation by altering one’s view of work in a more personally meaningful way [78,117]. Cognitive crafting was proved to be an interesting cognitive strategy to stay engaged for health care workers, a strategy which acts through the altering of the perceptions one has about professional tasks and relationships [119]. Since cognitive crafting does not change objective characteristics of the job but, rather, takes place inside the mind of the employee, as mental strategy, in contrast to relational and task crafting, it is considered that the aim is to enhance the meaningfulness of work [119]. Moreover, cognitive crafting can be considered the first step in the job crafting process because all job crafting initiatives start with cognitive evaluation of the characteristics of the job and person–job fit [120].

Regarding the relationship between cognitive crafting on one side and optimism, resilience, and self-efficacy on the other, as mediators in the proposed mediation chain, studies show that job crafting had a positive relationship with future personal resources, such as hope, resilience, self-efficacy, and optimism [79], and buffered the impact of job demands on job strain through job and personal resources [1]. Based on the COR theory’s main statement that resources do not exist in isolation, instead they reinforce each other and build resource caravans [29], we hypothesized that cognitive crafting buffers the impact of role ambiguity on dentists’ burnout and proposed the following hypotheses:

**Hypothesis** **5** **(H5).***Cognitive crafting is positively related to resilience, optimism, and self-efficacy*.

**Hypothesis** **6** **(H6).***Role ambiguity is negatively and indirectly related to burnout, mediated by cognitive crafting and resilience*.

**Hypothesis** **7** **(H7).***Role ambiguity is negatively and indirectly related to burnout, mediated by cognitive crafting and optimism*.

**Hypothesis** **8** **(H8).***Role ambiguity is negatively and indirectly related to burnout, mediated by cognitive crafting and self-efficacy*.

Figure 1, Figure 2 and Figure 3 summarize the hypothesized models.

## 2. Materials and Methods

### 2.1. Sampling and Participants

The main inclusion criterion was that all dentists were in practice with a legally required qualification. Starting from a list of active medical offices in the two north-west Romanian counties, the dentists were contacted by phone and asked about their willingness to participate in the study. Those who answered and expressed their availability were sent, via email or phone, the link to complete an online questionnaire. Consequently, the sample for the current research was reduced to a geographical area and the generality of the results was affected. On the other hand, this selection technique allowed for control regarding the targeted characteristics of the respondents. Future replications of the study should include other geographical areas to increase the generalization power of the results. We used a cross-sectional design to test the hypotheses. The survey was conducted from January 2022 to March 2022, when we decided to start statistical analyses on the respondents, composed of 191 active dentists. The age of the respondents ranged from 23 to 64 years of age (M = 34.93, SD = 8.64). Respondents’ length of service ranged from 1 year to 39 years (M = 9.42 years, SD = 8.64). Other socio-demographic details of the sample are presented in Table 1.

### 2.2. Measures

Participants completed an online questionnaire that covered role ambiguity as organizational stressor, the mediators of cognitive crafting, resilience, optimism, and self-efficacy, emotional exhaustion as burnout measure, and demographic characteristics. Two scales (*Role ambiguity* scale and *Cognitive crafting* scale) were adapted from previous relevant studies and analyzed in terms of construct validity on the present sample. These two instruments were translated from English to Romanian by one translator and the back translation (from Romanian to English) was realized independently by another translator. The other instruments were already validated in the Romanian population. To assess construct validity of the measures in the current sample, we conducted confirmatory factor analysis (CFA) using the Lavaan package [121] in R [122]. Cronbach’s alpha values for all scales were adequate for the present sample and are presented in Table 2. The detailed descriptions of each scale, regarding reliability, validity, and their use in previous studies, are further presented.

*Emotional exhaustion* was evaluated based on the Maslach Burnout Inventory (MBI; [123]), an instrument designed to assess the three components of burnout syndrome: emotional exhaustion (EE), depersonalization (DP), and reduced personal accomplishment (PA). The instrument was previously validated among 1190 Romanian healthcare professionals [124]. Data provided evidence to support the hypothesized three-factor model for the Romanian population and confirmed the invariance of the model across professional role, gender, age, and organizational tenure with adequate fit indicators: χ^2^ (86) = 432.29, CFI = 0.94, GFI = 0.95, NFI = 0.93, and RMSEA = 0.05 [124]. The dimension with the highest factor loading was emotional exhaustion, ranging between 0.72 and 0.85. Cynicism had the lowest factor loading, from 0.43 to 0.65 [124]. The instrument was previously used in different studies including Romanian healthcare professionals [58,125,126]. Starting from the fact that emotional exhaustion is approached as the core construct of burnout [14,28,52,56,58]) we used only the scale of Emotional exhaustion for measuring burnout in the present research. The scale had 5 items (e.g., “I feel emotionally drained from my work.”). Respondents were asked to evaluate the items regarding the frequency with which they experienced the described feelings on a seven-point scale from 0 (never) to 6 (always). In terms of construct validity, confirmatory factor analysis (CFA) using R [122] indicated very good fit measures [χ^2^ = 8.174, df = 5, *p* < 0.147, CFI = 995, TLI = 0.991, RMSEA = 0.058, SRMR = 0.017] for the presented sample. The scale of emotional exhaustion had excellent reliability measured through internal consistency (α = 0.91) in the current sample.

*Role ambiguity* was measured with a scale from a multi-dimensional instrument proposed by Tate and collaborators [127]. The authors reported adequate reliability coefficients for each sub-scale [127]. Similarly, the present study found adequate internal consistencies for the role ambiguity scale (α = 0.84), as well as very good fit measures [χ^2^ = 247.62, df = 3, *p* < 0.001, CFI = 1, TLI = 1, RMSEA = 0, SRMR = 0] in terms of construct validity. The scale was used in previous research [128,129,130]) and it has exhibited good psychometric properties. A sample item for the role ambiguity scale which contained 3 items was:” It is clear to me what others expect of me in my job”). Participants had to rate each item on a five-point Likert scale (1 = strongly disagree to 5 = strongly agree). For all the items the reversed score was calculated.

*Cognitive crafting* was measured using the five-item cognitive crafting scale from the Job Crafting Questionnaire (JCQ) [117]. Reliability analyses conducted by the authors indicated that the instrument had high internal consistency (0.89) [117]. Regarding construct validity, both EFA and CFA analyses revealed a three-factor structure [117] and a large body of research previously used this instrument (i.e., [131,132,133]). Following the idea that cognitive crafting is perhaps the facet of job crafting that aligns most closely to “work identity”, which is essentially how people define or perceive themselves at work [78,134], we used only the cognitive crafting scale of JCQ for the current research. Our CFA analysis, in terms of construct validity, indicated very good fit measures [χ^2^ = 608.887, df = 10, *p* < 0.001, CFI = 0.953, TLI = 0.906; RMSEA = 0.172, SRMR = 0.004] for the present research. The internal consistency in the current sample was high (α = 0.90). An example of an item was: “Think about how my job gives my life purpose.” Participants were instructed to indicate the extent to which they engaged in each crafting using a Likert-type scale (1 = “never”, 5 = “always”).

*Resilience* was measured using the items which assessed the dimension of resilience from The Psychological Capital Questionnaire (PCQ) proposed by Luthans and collaborators [75]. The PsyCap Questionnaire comprises four subscales (self-efficacy, hope, resilience, and optimism) and was previously validated in Romania [135]. Adequate Cronbach’s alpha values for each of the subscales were reported in the Romanian population (0.90 for self-efficacy scale; 0.84 for hope scale; 0.72 for resilience scale and 0.77 for optimism scale) [135]. There were six specific questions used to measure resilience (e.g., “I usually take stressful things at work in my stride”). Each item was rated on a 6-point Likert scale from 1 = strongly disagree to 6 = strongly agree. In terms of construct validity in the current sample, CFA analysis indicated very good fit measures [χ^2^ = 25.003, df = 9, *p* < 0.005, CFI = 0.957, TLI = 0.929, RMSEA = 0.096, SRMR = 0.049], as well as an adequate value for the internal consistency (α = 0.76).

*Optimism* was measured using the six items scale which assessed the dimension of optimism in The Psychological Capital Questionnaire (PCQ) [75] presented above, and used in research among different professional categories (i.e., [136,137]). The participants rated each item on a six-point Likert scale from 1 = strongly disagree to 6 = strongly agree. An example of an item was: “When things are uncertain for me at work, I usually expect the best”. The internal consistency in the current sample was low (α = 0.66) but CFA analysis indicated good fit measures [χ^2^ = 40.117, df = 9, *p* < 0.001, CFI = 0.915, TLI = 0.859, RMSEA = 0.135, SRMR = 0.082].

*Self-efficacy* was measured with the Generalized Self-Efficacy Scale [138] which was used in numerous studies, with adaptations for 33 languages, including Romanian [139]. According to the authors, the SES scale exhibited good psychometric properties, with high Cronbach-alpha values [140]. The scale contains 10 items. Possible responses were scored on a 4-point Likert scale, as follows: not at all true (1), hardly true (2), moderately true (3). and exactly true (4). An example of an item was: “I can solve most problems if I invest the necessary effort”. For the present sample we reported excellent internal consistencies (α = 0.91), as well as very good fit measures [χ^2^ = 107.389, df = 35, *p* < 0.001, CFI = 0.937, TLI = 0.919, RMSEA = 0.104, SRMR = 0.048] in terms of construct validity.

### 2.3. Data Collection

Data were collected using Google Forms through a single administration of a series of psychometric instruments to measure the participants’ perceptions of their working conditions and personal characteristics. The necessary time for filling the survey (between 15 and 20 min) was presented on the first page. A single link was used for all respondents to complete the online survey. For an anonymous survey approach, we followed specific recommendations to ensure that there were no multiple or malicious responses [141]. Thus, when surveys were completed, we looked at how quickly they were completed. We also looked at variability in responses to rating scale questions and the content of text responses. We did not identify questionnaires completed much more quickly than legitimate surveys, missing text response or little variability in rating scale responses (i.e., all being “Strongly Disagree”). No missing data were recorded.

#### Ethical Considerations

The purpose of the study, the anonymity of the responses, and ethical aspects relevant to the informed consent were presented before the first section of questions. All participants completed written informed consent forms before taking part in the study. On the consent page, respondents were advised to leave the study at any time or not take part if they felt uncomfortable thinking about their personal characteristics or feelings. This study was conducted in accordance with the guidelines of the Declaration of Helsinki and approved by the Ethics Committee of the University of Oradea (protocol code 2906/11.10.2021).

### 2.4. Data Analyses

We assessed the possibility that an integrated model (model 4), with all variables included (role ambiguity, cognitive crafting, resilience, optimism, self-efficacy, and emotional exhaustion), was valid. We performed a path analysis without latent variables using the Lavaan package [121] in R [122]. We calculated two relative fit indices (TLI—Tucker–Lewis’s index and CFI—Comparative fit index), and three absolute fit indices (the chi-square statistic; SRMR—the standardized root mean square residual, and RMSEA—the root mean square error of approximation). The model fit indicators were very poor [χ^2^ = 184.065, df = 6, *p* < 0.001, CFI = 0.559, TLI = −0.101, RMSEA = 0.394, SRMR = 0.202]. Consequently, the decision to conduct a separate analysis of each of the three conceptual mediation modes was sustained. The analysis of the correlation between the study variables (role ambiguity, cognitive crafting, resilience, optimism, self-efficacy, and emotional exhaustion) was performed with the Statistical Package for Social Science (SPSS) v23 program (IBM Corp., Armonk, NY, USA). The Pearson correlation coefficients were examined. To identify and explain the relationship between the dependent variable Y (emotional exhaustion) and the independent variable X (role ambiguity), which might be affected by W (other variables acting in the chain of variables), as we assumed in the proposed mediation models, we conducted a path analysis to test the mediation effects, using PROCESS macro for the SPSS v23 program. We performed the bootstrapping procedure by [142], using one independent variable, two mediators, and one dependent variable. We calculated 95% confidence intervals (CIs), based on bias-corrected bootstrap analyses with 5000 repetitions to analyze indirect effects. In the present research, we tested the proposed mediation models considering two mediators in the chain. For the first model we analyzed cognitive crafting and resilience, which further affect emotional exhaustion after being impacted by role ambiguity. For the second model we analyzed cognitive crafting and optimism, which further affect emotional exhaustion after being impacted by role ambiguity. For the third model we analyzed cognitive crafting and self-efficacy, which further affect emotional exhaustion after being impacted by role ambiguity. No differences were recorded according to gender in terms of role ambiguity, cognitive crafting, resilience, optimism, self-efficacy, or emotional exhaustion. Moreover, age, gender, marital status, and length of work experience showed no correlation with emotional exhaustion as dependent variable for the sample presented. Thus, socio-demographic variables were not controlled when we tested the mediation models.

## 3. Results

Means, standard deviations, internal consistency, and correlations for the study’s variables are shown in Table 2.

The correlations between variables were in the expected directions with no exceptions. The data showed a negative correlation between role ambiguity, as independent variable on one side, and all mediators (*r* = −0.34, *p* < 0.01 with cognitive crafting; r = −0.39, *p* < 0.01 with resilience; r = −0.41, *p* < 0.01 with optimism; r = −0.43, *p* < 0.01 with self-efficacy) on the other side. Consistent with Hypothesis 1, which stated that role ambiguity is positively related to burnout, the correlation between the two variables was observed (*r* = 0.34, *p* < 0.01). Further, a positive correlation between cognitive crafting, as permanent mediator in all mediation models, and personal resources was observed (r = 0.36, *p* < 0.01 with resilience; r = 0.43, *p* < 0.01 with optimism; r = 0.52, *p* < 0.01 with self-efficacy), as well as a negative correlation between cognitive crafting and burnout (r = −0.42, *p* < 0.01). The results were in line with hypothesis 5, which stated that cognitive crafting is positively related to resilience, optimism, and self-efficacy. The results offered support for Hypotheses 2, 3, and 4, which stated that resilience (H2), optimism (H3), and self-resilience (H4) are negatively related to burnout (r = −0.43, *p* < 0.01 burnout with resilience; r = −0.53, *p* < 0.01 burnout with optimism; r = −0.44, *p* < 0.01 burnout with self-efficacy).

Going further with the testing of the mediation models, as depicted in Table 3, we first computed the model for cognitive crafting as the first mediator (M1) and for resilience as the second mediator (M2) in the mediation chain. According to our results, role ambiguity was significantly related to cognitive crafting (b = −0.78, *p* < 0.001), and a high level of cognitive crafting predicted a high level of resilience (b = 0.31, *p* < 0.001). Next, we computed the analyses for resilience (M2) as the second mediator, and, as was stated, resilience was strongly and negatively related to emotional exhaustion (b = −0.34, *p* < 0.000). When we controlled the effect of mediators, role ambiguity was positively and significantly related to emotional exhaustion (b = 0.54, *p* < 0.03) and this direct effect was smaller than the total effect (b = 1.28, *p* < 0.001).

Consistent with Hypothesis 6, role ambiguity was negatively and indirectly related to burnout, mediated by cognitive crafting and resilience. The sequential indirect effect of role ambiguity on emotional exhaustion, via both mediators in the series (cognitive crafting and resilience), was significant, positive, and smaller than the direct effect (b = 0.08, 95% CI [0.021, 0.193]). Moreover, there was also a positive indirect effect of role ambiguity on emotional exhaustion through cognitive crafting (b = 0.35, 95% CI [0.182, 0.568]) and resilience (b = 0.29, 95% CI [0.100, 0.577]). Overall, there was partial mediation support for the proposed theoretical model. Thus, when acting in a chain, cognitive crafting, and resilience buffered the negative impact of role ambiguity on emotional exhaustion.

In line with Hypothesis 7, role ambiguity was indirectly related to burnout, totally mediated by cognitive crafting and optimism. As depicted in Table 4, the sequential indirect effect of role ambiguity on emotional exhaustion, via both mediators in the series (cognitive crafting and optimism), was positive, significant, and smaller than the total effect (b = 0.16, 95% CI [0.078, 0.294]). In addition, an indirect effect of role ambiguity on emotional exhaustion only through cognitive crafting (b = 0.87, 95% CI [0.572, 1.248]), and then, only through optimism (b = 0.42, 95% CI [0.217, 0.704]), was noticed. More importantly, the direct effect of role ambiguity on burnout was statistically insignificant (b = 0.41, 95% CI [−0.070, 0.896]). This result confirmed that there was total mediation support for the proposed theoretical model, which meant that cognitive crafting and optimism reinforced each other and completely buffered the negative impact of role ambiguity on emotional exhaustion.

Furthermore, the results sustained Hypothesis 8, which stated that the role of ambiguity is indirectly related to burnout, mediated by cognitive crafting and self-efficacy. A negative indirect effect of role ambiguity on emotional exhaustion. only through cognitive crafting (b = 0.30, 95% CI [0.124, 0.518]), and only through self-efficacy (b = 0.26, 95% CI [0.111, 0.500]), was noticed. However, when both mediators were analyzed in chain, the relation of mediation was statistically significant (b = 0.13, 95% CI [0.050, 0.269]). The decrease of the coefficients value when analyzing the indirect chain mediation effect compared to the direct effect, or to the total effect, confirmed that there was a partial mediation support for the third theoretical model, which assumed that cognitive crafting and self-efficacy buffered the negative impact of role ambiguity on emotional exhaustion in their sequential action. The results are presented in Table 5.

For simplicity, the values in the various paths for conceptual models as pictorial representations are presented below in Figure 4, Figure 5 and Figure 6.

## 4. Discussion

This study proposed three conceptual models which integrated assumptions from JD-R, COR, and self-regulation theories. The aim was to examine the mediating role of cognitive crafting and personal resources (resilience, optimism, and self-efficacy) in the association between Romanian dentists’ role ambiguity and emotional exhaustion. The results sustained the validity of the advanced mediation models and the theoretical assumptions on which they were built. Thus, the JD-R model was substantiated in its claim that burnout comes from an imbalance between job demands and resources [1] and that personal resources are expected to function as a buffer between high job demands and undesirable health consequences, such as burnout [12]. Accordingly, our findings verified that role ambiguity, as job demand, increased emotional exhaustion, whereas personal resources decreased it. Moreover, resilience, optimism, and self-efficacy diminished burnout both when were analyzed as single mediators and then when they were added in the mediation chain after cognitive crafting. The results were in the direction assumed by the self-regulation theories because cognitive crafting reduced burnout when it was analyzed as a single mediator. More importantly, we noticed that cognitive crafting reduced the role ambiguity effect on emotional exhaustion, because the biggest effect on emotional exhaustion depletion was observed when cognitive crafting was included in the serial mediation as first mediator, before personal resources. We explained the result as being due to proactive initiatives related to cognitive crafting increasing personal resources. In other words, cognitive crafting reinforced resilience, self-efficacy, and optimism in a mediation chain through the effect of the resource caravan [29], an effect reported by previous research [23]. Therefore, the results sustained the main assumption of the COR theory, which is that resources do not exist in isolation but reinforce each other [29]. Furthermore, our findings supported a recent approach of the JD-R model, that the engagement in job crafting behavior leads to higher levels of job and personal resources [36]. Thus, our results were in line with previous research regarding the role of job crafting in the decreasing of employee burnout [17,18,19].

However, the striking result in the current research was that the mediation chains buffered, partially or totally, the impact of role ambiguity on burnout. Thus, after previously observing a strong negative effect of role ambiguity on emotional exhaustion in the absence of the control for cognitive crafting, resilience, optimism, and self-efficacy, this effect became null or smaller, depending on the mediation model. Specifically, for the mediation chains proposed in the first and in the third model (cognitive crafting followed by resilience, and cognitive crafting followed by self-efficacy, respectively) we recorded a decrease in emotional exhaustion because of a partial mediation effect. In the case of the mediation chain formed by cognitive crafting and optimism, which were proposed in the second model, the effect of role ambiguity on cognitive crafting disappeared and a total mediation effect was reported. Our results completed previous research regarding the role of resilience, optimism, and self-efficacy in reducing burnout [27,73,85,87,88,95], or in buffering the distress from job demands on burnout [74]. In accordance with other studies [1,104] our results showed that job crafting had a positive relationship with future personal resources and buffered the impact of job demands on job strain through job and personal resources. Consequently, we observed that cognitive crafting and personal resources acted as a buffer when included as serial mediators between role ambiguity and emotional exhaustion. Based on previous research and considering the findings of this study, we appreciated that personal resources could have the same function as job resources, especially when other resources, such as appealing to cognitive crafting, are available. The emergence of resilience, optimism, and self-efficacy, through theory significant indirect role in diminishing emotional exhaustion, recorded by the present research further demonstrates the importance of personal resources in promoting dentists’ wellbeing. In addition, the proved role of cognitive crafting as an efficient mediator in emotional exhaustion reduction indicates that it can, therefore, be considered an important factor in both the prevention of, and treatment of, symptoms of dentists’ burnout.

### 4.1. Practical Implications

It is known that cognitive crafting strategies, resilience, optimism, and self-efficacy can be taught and learned [92,101,143,144,145,146]. This approach opens the possibility to understand cognitive crafting and personal resources not only as personal but, moreover, as professional skills which might be learnt. Consequently, we support the necessity for development strategies of these individual variables which buffer burnout in the case of dentists, who are frequently prone to emotional exhaustion [9]. Since these variables involve cognitive processing because cognitive crafting is considered a mental strategy [119], resilient responses are promoted by the situationally appropriate application of flexibility or rigidity in affective–cognitive systems [81]. Optimism and self-efficacy involve a cognitive dimension [75,100]. We suggest that their enhancement can be served by cognitive behavioral therapy, the effectiveness of which has been scientifically proven [147]. Moreover, the literature has identified different cognitive crafting strategies, like reminders of the holistic purpose of a job, rather than perceiving the job as a set of distinct tasks and relationships, making mental linkages between specific tasks, on one side, and personal interests that are meaningful to the employee, on the other side, or actively focusing on the aspects of work that are most meaningful to the employee [148]. Thus, dentists could frame their job as improving their client’s self-esteem, rather than just solving a specific medical problem, as an example of expanding perceptions for the first mentioned cognitive coping strategy [148]. Going further, a dentist with a hobby for painting could make a mental link between the task of finishing a tooth and the action of creating a masterpiece. This cognitive crafting strategy involves linking perceptions [148]. Finally, dentists could actively focus on the aspects of work that are the most significant to them, a strategy also called focusing perceptions [148]. For example, a dentist who dislikes explaining to patients all the necessary procedures for the best result, but perceives challenging intervention as an enjoyable task, might reframe the communication with the client to a self-preparation plan, indispensable for the success of the intervention. Furthermore, a short and accessible training procedure, known as “micro-intervention”, the Psychological Capital Intervention (PCI, [149]), was validated in a different cultural context [150] for enhancing resilience, optimism, or self-efficacy. Thus, practical implications are related to efficient professionals or self-help interventions which reduce emotional exhaustion.

### 4.2. Limitations and Future Research

There are several limitations in the present research which must be underlined. First, the present study was exclusively based on self-reported data. It would be beneficial for future research to incorporate observations or clinical interviews especially for burnout symptomatology and qualitative methodology for identifying cognitive crafting strategy and the frequency of its use. Second, we conceptualized and operationalized the research using a cross-sectional design. Consequently, our research cannot assess if there would be a change in variables over time. The relations found do not involve causal inferences between the studied variables. Regarding causality, we cannot be sure that cognitive crafting causes optimism, self-efficacy or resilience which further causes low emotional exhaustion. Longitudinal studies could further strengthen our conclusions and provide evidence of the nature of the relationships between role ambiguity and emotional exhaustion. Moreover, the number of participants limited interpretation of the results. Additional studies should use larger samples to confirm our findings. Last, but not least, possible confounding factors, like socio-demographic aspects, might influence the relationship between the investigated variables. Although our prior analysis did not show differences between gender in terms of the studied variables, or significant correlations between age, gender, length of work experience or marital status and emotional exhaustion, a statistical control for socio-demographic factors is recommended. Future research should also test other job demands and mediators in relation to burnout to provide a broader image of the psychological implications in preventing dentists’ emotional exhaustion.

In sum, our results provide a more in-depth insight into the nature of relationships between role ambiguity and emotional exhaustion reinforcing the fact that cognitive crafting and personal resources reduce burnout, based on evidence from a less explored occupational group and regional context. This study reinforces the idea that emotional exhaustion antecedents deserve much more research and analysis related to different occupations.

## 5. Conclusions

The research objective was fulfilled because the results highlighted that the inclusion of personal resources, as a proximal outcome of job crafting, further explained the link between role ambiguity and emotional exhaustion in the case of Romanian dentists. Thus, it can be assumed that adequate resources available to employees allow them to cope with job demands, which further leads to reduction, or even nullification, of the risk of burnout.

## Figures and Tables

**Figure 1 ijerph-19-16617-f001:**
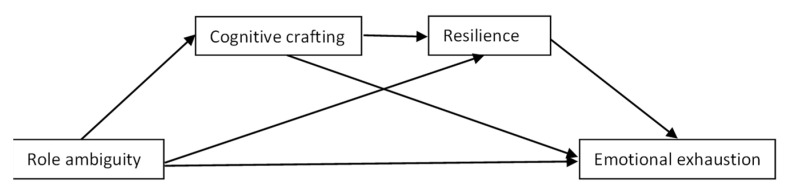
Hypothetical chain mediation model 1 (serial mediators: cognitive crafting and resilience).

**Figure 2 ijerph-19-16617-f002:**
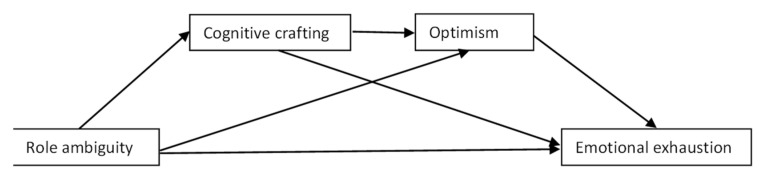
Hypothetical chain mediation model 2 (serial mediators: cognitive crafting and optimism).

**Figure 3 ijerph-19-16617-f003:**
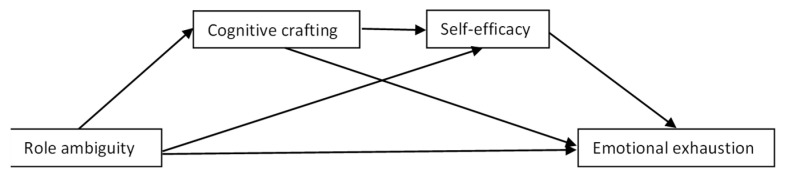
Hypothetical chain mediation model 3 (serial mediators: cognitive crafting and self-efficacy).

**Figure 4 ijerph-19-16617-f004:**
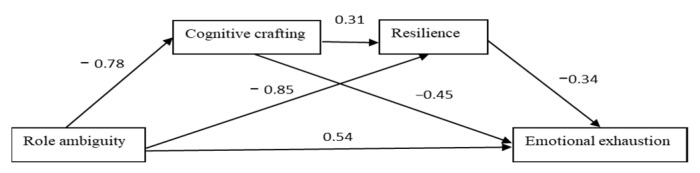
Paths values for the sequential mediation model 1.

**Figure 5 ijerph-19-16617-f005:**
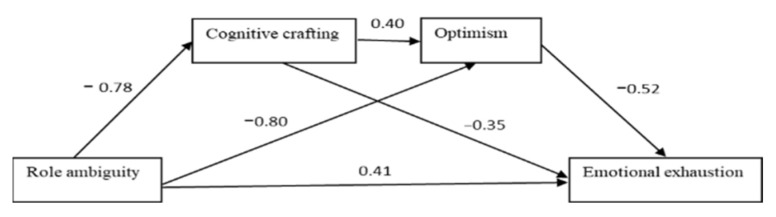
Paths values for the sequential mediation model 2.

**Figure 6 ijerph-19-16617-f006:**
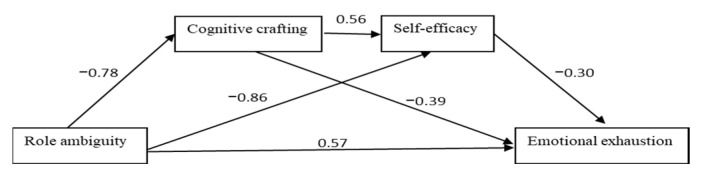
Paths values for the sequential mediation model 3.

**Table 1 ijerph-19-16617-t001:** The demographic profile of respondents (N = 191).

	Frequency	Percent
*gender category*		
Male	49	25.7
Female	142	74.3
*marital status*		
Single	45	23.6
married/in a relationship	146	76.4
*length of service*		
less than 5 years	70	36.6
between 6–10 years	56	29.3
between 11–20 years	40	20.9
21 years and above	25	13.2

**Table 2 ijerph-19-16617-t002:** Means, standard deviations and correlation coefficients between variables (N = 191).

Variables	M	Sd	1.	2.	3.	4.	5.	6.
1. Role ambiguity	4.04	1.69	(0.84)					
2. Cognitive crafting	21.37	3.84	−0.34 **	(0.90)				
3. Resilience	27.73	4.74	−0.39 **	0.36 **	(0.76)			
4. Optimism	27.43	4.66	−0.41 **	0.43 **	0.65 **	(0.66)		
5. Self-efficacy	34.29	5.16	−0.43 **	0.52 **	0.61 **	0.58 **	(0.91)	
6. Burnout	13.63	6.26	0.34 **	−0.42 **	−0.420 **	−0.53 **	−0.44 **	(0.91)

Notes: ** *p* < 0.01, one single tailed. Internal consistency alphas are displayed in the diagonal.

**Table 3 ijerph-19-16617-t003:** Total, direct, and indirect effects of the sequential mediation model 1—role ambiguity on burnout through cognitive crafting and resilience (PROCESS).

Variables	Unst. Coef.	SE	*p*	BC Bootstrap 95% CI	
				Lower	Upper
The total effect of					
role ambiguity→burnout	1.28	0.25	0.001	0.790	1.781
The direct effect of:					
role ambiguity→burnout (c’)	0.57	0.25	0.02	0.063	1.082
role ambiguity→cognitive crafting	−0.78	0.15	0.001	−1.089	−0.479
role ambiguity→self-efficacy	−0.86	0.19	0.001	−1.244	−0.490
cognitive crafting→burnout	−0.39	0.12	0.001	−0.630	−0.154
cognitive crafting→self-efficacy	0.56	0.08	0.001	0.402	0.735
self-efficacy→burnout	−0.30	0.09	0.001	−0.492	−0.123
The indirect effect of:					
role ambiguity→cognitive crafting→burnout	0.30			0.124	0.518
role ambiguity→self-efficacy→burnout	0.26			0.111	0.500
role ambiguity→cognitive crafting→self-efficacy→burnout	0.13			0.050	0.269

Note: *n* = 191; number of bootstrap samples for percentile bootstrap confidence intervals: 10,000.

**Table 4 ijerph-19-16617-t004:** Total, direct, and indirect effects of the mediation model 2—role ambiguity on burnout through cognitive crafting and optimism (PROCESS).

Variables	Unst. Coef.	SE	*p*	BC Bootstrap 95% CI	
				Lower	Upper
The total effect of					
role ambiguity→burnout	1.28	0.25	0.001	0.790	1.781
The direct effect of:					
role ambiguity→burnout (c’)	0.41	0.24	0.09	−0.070	0.896
role ambiguity→cognitive crafting	−0.78	0.15	0.001	−1.089	−0.479
role ambiguity→optimism	−0.80	0.18	0.001	−1.169	−0.449
cognitive crafting→burnout	−0.35	0.10	0.001	−0.569	−0.137
cognitive crafting→optimism	0.40	0.08	0.001	0.247	0.565
optimism→burnout	−0.52	0.09	0.001	−0.711	−0.344
The indirect effect of:					
role ambiguity→cognitive crafting→burnout	0.87			0.572	1.248
role ambiguity→optimism→burnout	0.42			0.217	0.704
role ambiguity→cognitive crafting→optimism→burnout	0.16			0.078	0.294

Note: *n* = 191; number of bootstrap samples for percentile bootstrap confidence intervals: 10,000.

**Table 5 ijerph-19-16617-t005:** Total, direct, and indirect effects of the mediation model 3—role ambiguity on burnout through cognitive crafting and self-efficacy (PROCESS).

Variables	Unst. Coef.	SE	*p*	BC Bootstrap 95% CI	
				Lower	Upper
The total effect of					
role ambiguity→burnout	1.28	0.25	0.001	0.790	1.781
The direct effect of:					
role ambiguity→burnout (c’)	0.57	0.25	0.02	0.063	1.082
role ambiguity→cognitive crafting	−0.78	0.15	0.001	−1.089	−0.479
role ambiguity →self-efficacy	−0.86	0.19	0.001	−1.244	−0.490
cognitive crafting→burnout	−0.39	0.12	0.001	−0.630	−0.154
cognitive crafting→self-efficacy	0.56	0.08	0.001	0.402	0.735
self-efficacy→burnout	−0.30	0.09	0.001	−0.492	−0.0123
The indirect effect of:					
role ambiguity→cognitive crafting→burnout	0.30			0.124	0.518
role ambiguity→self-efficacy→burnout	0.26			0.111	0.500
role ambiguity→cognitive crafting→self-efficacy→burnout	0.13			0.050	0.269

Note: *n* = 191; number of bootstrap samples for percentile bootstrap confidence intervals: 10,000.

## Data Availability

Not applicable.

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
