# Peer review of "The Mediation Chain Effect of Cognitive Crafting and Personal Resources on the Relationship between Role Ambiguity and Dentists’ Emotional Exhaustion"

_ijerph, 2022, doi:10.3390/ijerph192416617_

Round 1

Reviewer 1 Report

General comments

This is an informative article that uses JD-R, COR and self-regulation theories integration to explain the relationship between role ambiguity and dentists' burnout, treating cognitive crafting and personal resources (such as resilience, optimism and self-efficacy) as joint/chain mediators in the association. The article addresses a major gap in knowledge and also focuses on an underrepresented population in Romania. The major strength of this work is the author's attention to detail in the review of related literature, whereas its major weakness is in the methodology. The authors need to proofread or have a colleague proofread the article for several grammatical errors. After reviewing the manuscript, I recommend major revisions with the following comments:

Abstract

1. Provide the full meaning of these abbreviations (JD-R, COR) before you can subsequently abbreviate. See the attached file for some other minor and specific errors.

Introduction

2. Why were all the personal resources not incorporated into one model? It could offer other insights if a fourth mediation model is added, where all the personal resources (mediators) could be chained in linking role ambiguity to dentists' burnout.

3. Why did you focus on just the "emotional exhaustion" dimension as the criterion variable or a proxy for job burnout in this study? In fact, why were other dimensions - depersonalisation, and reduced personal accomplishment, not included in the study? Although the authors wrote, “Previous research already showed that the dental profession is generally linked to high levels of burnout syndrome (Gómez-Polo et al., 2022; Ozarslan & Caliskan, 2021; Singh et al., 2016) mainly in the subscale of emotional exhaustion (da Silva Moro et al., 2022; Slabšinskiene et al., 2021). This is the reason why we choose for the present study to measure only the emotional exhaustion if the manifestations like constant fatigue or feelings of lacking strength are the common elements for every dentist as the first phase of burnout (Maslach, 1996).” The reason advance does not provide enough justification for ignoring the other dimensions of burnout. Therefore, the authors need to provide a strong justification for choosing only the emotional exhaustion dimension and ignoring the other two.

Methods

4. Why was convenience sampling used instead of probability methods, which have a greater generalisation power? Also, be explicit about your selection/sampling process to allow for replication (See the attached pdf file for other comments).

5. Explain your snowballing process.

6. Which survey did you use? Kindly provide a detailed description. The more reason you should place the measures section ahead of the data collection section. We need to understand your measures first before knowing how data were collected.

7. Separate the data collection process from ethical considerations by creating a sub-heading for the latter. Talk about the ethical review, potential risks, informed consent and anonymity, among other ethics. Be thorough about your methodology.

8. For data collection, explain the snowballing stages, including how it started, expanded and ended. Explain why you stopped accepting responses at the 191st respondent. How did you ensure that there were no multiple and/or malicious responses aimed at skewing your results?

9. You have done well by describing these scales and providing sample items. However, where is the validity and reliability information of the scales from the original developers and other researchers that have previously used them?

10. Did the authors also consider revalidating the scales on their population? If yes, where is the validity and reliability information; if no, why was further validity and reliability of the scales not done since they were developed in other contexts different from where the current study was conducted? If the scales are not valid and suitable for your respondents, then the results so obtained will be spurious and misleading.

Results

11. The authors have provided the results of the analysis in Tables (which is fine). However, I was expecting to see the actual fitting of the chain models in Figures 1, 2 and 3 in the results section. Conceptual models need to be tested by fixing the values in the various paths for pictorial representation. Thus, we should have a total of six figures in this work.

12. All p < .000 should be reported as p < .001.

Author Response

Dear Sir/Madamme,

Thank you for your comments and suggestions. I have appreciated all of them as being extremely valuable and hope I have included all your observations as follow:

Abstract

  1. Provide the full meaning of these abbreviations (JD-R, COR) before you can subsequently abbreviate. See the attached file for some other minor and specific errors.

 - Response: Thank you for your comment. I have provided the full meaning of JD-R and COR abbreviation and I have corrected all specific editing errors. Also, I have asked for a proofreading to detect and to correct other syntax errors.

Introduction

  1. Why were all the personal resources not incorporated into one model? It could offer other insights if a fourth mediation model is added, where all the personal resources (mediators) could be chained in linking role ambiguity to dentists' burnout.

- Response: Thank you for your comment. I have tested the fourth mediation model because I was interested from the beginning about the explicative power of an integrated model but the results of the path analysis with sequential and parallel mediators provided very poor indicators for this model. I insert in the attachment the path diagram of the model and all path coefficients.

Consequently, I have appreciated this integrated model as inadequate, and I have considered that the separate analyses for the models are proper. However, taking your suggestion into consideration, I have included the following paragraph in the article: ” We assess the possibility that an integrated model (M4) with all variables included (role ambiguity, cognitive crafting, resilience, optimism, self-efficacy, and emotional exhaustion) to be valid. We perform a path analysis without latent variable using the lavaan package (Rosseel, 2012) in R (R Core Team, 2021). We calculated two relative fit indices (TLI - Tucker-Lewis’s index and CFI - Comparative fit index), and three absolute fit indices (the chi-square statistic; SRMR - the standardized root mean square residual, and RMSEA - the root mean square error of approximation). The model fit indicators are very poor [χ2= 184.065, df = 6, p < .001, CFI = .559, TLI = -.101, RMSEA = .394, SRMR = .202]. Consequently, the decision for a separate analysis of each of the three conceptual mediation modes is sustained.

  1. Why did you focus on just the "emotional exhaustion" dimension as the criterion variable or a proxy for job burnout in this study? In fact, why were other dimensions - depersonalisation, and reduced personal accomplishment, not included in the study? Although the authors wrote, “Previous research already showed that the dental profession is generally linked to high levels of burnout syndrome (Gómez-Polo et al., 2022; Ozarslan & Caliskan, 2021; Singh et al., 2016) mainly in the subscale of emotional exhaustion (da Silva Moro et al., 2022; Slabšinskiene et al., 2021). This is the reason why we choose for the present study to measure only the emotional exhaustion if the manifestations like constant fatigue or feelings of lacking strength are the common elements for every dentist as the first phase of burnout (Maslach, 1996).” The reason advance does not provide enough justification for ignoring the other dimensions of burnout. Therefore, the authors need to provide a strong justification for choosing only the emotional exhaustion dimension and ignoring the other two.

 - Response: Thank you for your suggestion. I have revised this part accordingly, and I have included the following paragraph as a better justification for choosing only the emotional exhaustion dimension and ignoring depersonalization, and reduced personal accomplishment (line 155-173): When referring at the burnout division into three dimensions (emotional exhaustion, depersonalization, and reduced personal accomplishment/ professional efficacy), emotional exhaustion is viewed as the core component of burnout while depersonalization and reduced personal accomplishment are additional components (Brenninkmeyer et al., 2001; Maslach et al., 2001). It is argued that exhaustion as a form of strain, has a separate role from cynicism and personal accomplishment in burnout [Lee et al., 1996]. Furthermore, while cynicism is seen as a form of defensive coping in the burnout phenomenon, person-al accomplishment is considered a form of self-evaluation related to performance [Lee et al., 1996; Leiter, 1993], rather reflecting the employees’ personality and not their reactions to stressful situations (Koeske and Koeske, 1989). Moreover, in defining burnout, Pines et al. (1992) refer to the state of emotional, but also physical and mental exhaustion in long-lasting emotionally demanding situations.  Considering this aspect, Seidler et al. (2014) argue that it is becoming apparent that especially emotional exhaustion can be understood as the core component of burnout and conduct a systematic review which point to a relationship between psycho-social working conditions and the development of emotional exhaustion as the core dimension of burnout (Seidler et al., 2014). Other research conducted on Romanian healthcare professional measure emotional exhaustion as the representative component of burnout (i.e., Secosan et al., 2020).

Methods

  1. Why was convenience sampling used instead of probability methods, which have a greater generalisation power? Also, be explicit about your selection/sampling process to allow for replication (See the attached pdf file for other comments).

- Response: Thank you for your comment. I have described the sampling process and I have presented some pro and cons for the sampling method by including the following paragraph: „ The main inclusion criteria were that all dentists to be in practice with a legally required qualification. Starting from the list of active medical offices from the two north-west Romanian counties, the dentists were contacted by phone and asked about their willingness to participate in the study. Those who answered and expressed their availability were sent by email or by phone the link to complete the questionnaires. Thus, the sample for the current research is reduced to a geographical area and the generality of the results is affected. On the other side, this selection technique allows for control regarding the targeted characteristics of the respondents. Future replications of the study should include other geographical areas to increase the results generalization power. We used a cross-sectional design to test the hypotheses. The survey was shared between January and March 2022 when we decided to start statistical analyses. The research was conducted on a convenience sample composed of 191 active dentists.

  1. Explain your snowballing process.

- Response: Thank you for your comment. Although I have asked potential responders to spread the link for questionnaire to their dentist colleagues (this was an attempt to combine convenience sampling with probabilistic sampling) I am not sure about their involvement. Consequently, I have decided to give-up referring at the snowballing process and I have described sampling methodology without reference at sample type.

  1. Which survey did you use? Kindly provide a detailed description. The more reason you should place the measures section ahead of the data collection section. We need to understand your measures first before knowing how data were collected.

- Response: Thank you for your suggestion. I have included the following paragraph: ,, Participants completed an online questionnaire that covered role ambiguity as organizational stressor, cognitive crafting, resilience, optimism, self-efficacy as mediators, emotional exhaustion as burnout measure. Additionally, a set of socio-demographic data was collected, including information about gender, marital status, age, and work experience.

  1. Separate the data collection process from ethical considerations by creating a sub-heading for the latter. Talk about the ethical review, potential risks, informed consent and anonymity, among other ethics. Be thorough about your methodology.

- Response: Thank you for your comment. I have separated the data collection process from ethical considerations, and I have referred at anonymity and informed consent. Thus, I have included the following paragraph (line 454-460): ” The purpose of the study, the anonymity of the responses, and ethical aspects relevant to the informed consent was presented before the first section of questions. All participants completed written informed consent forms before taking part in the study. On the consent page, respondents were advised to quit at any time or not take part if they felt uncomfortable thinking about their personal characteristics or feelings. This study was conducted in accordance with the guidelines of the Declaration of Helsinki and approved by the Ethics Committee of the University of Oradea (protocol code 2906/11.10.2021).”

  1. For data collection, explain the snowballing stages, including how it started, expanded and ended. Explain why you stopped accepting responses at the 191st respondent. How did you ensure that there were no multiple and/or malicious responses aimed at skewing your results?

- Response: Thank you for your comment. I think I have solved the first part of this issue by answering at point 5 (Explain your snowballing process). Regarding the second part of the request, I have included the following paragraph: ” Data were collected using Google Forms through a single administration of a series of psychometric instruments to measure the participants’ perception of their working conditions and personal characteristics. The necessary time for filling the survey (between 15 and 20 min) was presented on the first page. A single link was used for all respondents to complete the online survey. For anonymous survey approach we follow specific recommendations to ensure that there were no multiple or malicious responses (Langford, 2020). Thus, when surveys were completed, we looked how quickly they were completed. We also looked at variability in responses to rating scale questions and the content of text responses. We did not identified questionnaires completed much more quickly than legitimate surveys, missing text response or showing little variability in rating scale responses (i.e., all being “Strongly Disagree”). No missing data were recorded.”

  1. You have done well by describing these scales and providing sample items. However, where is the validity and reliability information of the scales from the original developers and other researchers that have previously used them?

- Response: Thank you for your comment. I have included all requested information.

  1. Did the authors also consider revalidating the scales on their population? If yes, where is the validity and reliability information; if no, why was further validity and reliability of the scales not done since they were developed in other contexts different from where the current study was conducted? If the scales are not valid and suitable for your respondents, then the results so obtained will be spurious and misleading.

- Response: Thank you for your comment. I have performed a CFA in the current sample for all scale used as measures in the present research.

Results

  1. The authors have provided the results of the analysis in Tables (which is fine). However, I was expecting to see the actual fitting of the chain models in Figures 1, 2 and 3 in the results section. Conceptual models need to be tested by fixing the values in the various paths for pictorial representation. Thus, we should have a total of six figures in this work.

- Response: Thank you for your observation. I have included in the Results section the values of the paths for each model in the pictorial representation.

  1. All p < .000 should be reported as p < .001.

Response: Thank you. I have corrected.

Reviewer 2 Report

I believe that for this study, it would have been interesting to use the CESQT test (questionnaire for the evaluation of burnout syndrome at work) by PhD. Pedro Gil-Monte to measure burnout, because fom my experience I consider this is a scale that works very well and provides more information than the MBI. In this sense, the CESQT introduces the variables of enthusiasm for work, emotional exhaustion, indifference, guilt, and a total scale of burnout.

Regarding the conclusion, it is always interesting to return to the objective of investigation and explain if it has been fulfilled, likewise reviw the hypotheses to check if the initial assumptions are fullfilled and corroborate if they have been answered, and if they coincide with some studies that have been cited before.

Author Response

Dear Sir/Madame,

Please find below my response for each comment/request.

I believe that for this study, it would have been interesting to use the CESQT test (questionnaire for the evaluation of burnout syndrome at work) by PhD. Pedro Gil-Monte to measure burnout, because fom my experience I consider this is a scale that works very well and provides more information than the MBI. In this sense, the CESQT introduces the variables of enthusiasm for work, emotional exhaustion, indifference, guilt, and a total scale of burnout.

- Response: Thank you for your suggestion. I have two arguments for my decision:

  1. It is known that there are other established burnout questionnaires (i.e., Copenhagen Burnout Inventory, the Oldenburg Burnout Inventory, the Shirom Melamed Burnout Measure, Gillespie-Number of Burnout Inventory, the Tedium Measure), and of course the Questionnaire for the evaluation of occupational burnout syndrome (CESQT) but given the different definitions of burnout in these listed questionnaires, the comparability of the results is somewhat limited. For the present research I measured only emotional exhaustion, and I provided strong arguments based on other research using MBI which argue that emotional exhaustion is the core dimension of burnout (specific in case of MBI measuring). I do not know if this is the case for CESQT but I will take this instrument into consideration for my future research.
  2. An important issue when conducting research is to have the instruments validated on population you are working. To my knowledge, CESQT is not validated (yet) on Romanian sample. Instead, MBI had passed this process and is validated specific on healthcare professionals. Moreover, there are several articles which use MBI in case of Romanian medical staff. The articles I refer are:

Bria, M., Spânu, F., Băban, A., & DumitraÅŸcu, D. L. (2014). Maslach burnout inventory–general survey: factorial validity and invariance among Romanian healthcare professionals. Burnout Research, 1(3), 103-111.

Dimitriu, M. C., Pantea-Stoian, A., Smaranda, A. C., Nica, A. A., Carap, A. C., Constantin, V. D., ... & Socea, B. (2020). Burnout syndrome in Romanian medical residents in time of the COVID-19 pandemic. Medical hypotheses, 144, 109972.

Popa-Velea, O., Diaconescu, L. V., Gheorghe, I. R., Olariu, O., Panaitiu, I., Cerniţanu, M., ... & Nicov, I. (2019). Factors associated with burnout in medical academia: An exploratory analysis of Romanian and Moldavian physicians. International journal of environmental research and public health, 13(16).

Regarding the conclusion, it is always interesting to return to the objective of investigation and explain if it has been fulfilled, likewise reviw the hypotheses to check if the initial assumptions are fullfilled and corroborate if they have been answered, and if they coincide with some studies that have been cited before.

- Response: Thank you for your suggestion. I have linked the objective of investigation with the results in the Conclusion section.

Reviewer 3 Report

Dear authors,

Congratulations for your article, you have done a great work!

Few remarks and recommendations:

- at Table 1, the female are 74.3% (not 44.3%). It will be interesting if you add more factual data (age by interval, work experience by interval etc)

- I liked very much your data analysis, but... I have a problem with your hypothesis: H2 is quite a truism (high resilience involve low burn-out). Also, the same for optimism and self-efficacy. By other hand, H5 promote the connections between "cognitive crafting" and 3 different dependent variable (usually we use only one). If it is not to complicated, please try to adjust/explain somehow these issues.

- line 592, you have limited and limits into the same row, it is preferable to avoid this kind of repetition

- if you apply an online questionnaire, why is not available the database?

Congrats for your work, you got very interesting data!

Author Response

Dear Sir/Madame,

Please find below my response for each comment/request.

Few remarks and recommendations:

- at Table 1, the female are 74.3% (not 44.3%). It will be interesting if you add more factual data (age by interval, work experience by interval etc)

Response: Thank you for your observation. I have corrected the percentage and I have included age by interval, work experience by interval.

- I liked very much your data analysis, but... I have a problem with your hypothesis: H2 is quite a truism (high resilience involve low burn-out). Also, the same for optimism and self-efficacy. By other hand, H5 promote the connections between "cognitive crafting" and 3 different dependent variable (usually we use only one). If it is not to complicated, please try to adjust/explain somehow these issues.

Response: Thank you for your recommendation. Because when the aim of the study is to validate a mediation model (for the present research we have three models), the rule that is followed in research is to test each mediation path separately and to formulate the hypothesis accordingly. Moreover, as you noticed, the H2 is formulated as : ,,Resilience is negatively related to burnout,, not ” High resilience involve low burn-out.” The situation is the same for optimism and self-efficacy. I provide further some studies with a similar approach in hypothesis formulation:

Secosan I, Virga D, Crainiceanu ZP, Bratu T. The Mediating Role of Insomnia and Exhaustion in the Relationship between Secondary Traumatic Stress and Mental Health Complaints among Frontline Medical Staff during the COVID-19 Pandemic. Behav Sci (Basel). 2020 Oct 26;10(11):164. doi: 10.3390/bs10110164. PMID: 33114678; PMCID: PMC7692994.

Vîrgă, D.; Baciu, E.-L.; Lazăr, T.-A.; LupÈ™a, D. Psychological Capital Protects Social Workers from Burnout and Secondary Traumatic Stress. Sustainability 2020, 12, 2246. https://doi.org/10.3390/su12062246

Luceño-Moreno, Lourdes, et al. "Symptoms of posttraumatic stress, anxiety, depression, levels of resilience and burnout in Spanish health personnel during the COVID-19 pandemic." International journal of environmental research and public health 17.15 (2020): 5514.

- line 592, you have limited and limits into the same row, it is preferable to avoid this kind of repetition

Response: Thank you for your observation. I have reformulated.

- if you apply an online questionnaire, why is not available the database?

Response: Thank you for your suggestion. I will attach the database.